# Rangeland Biodiversity and Climate Variability: Supporting the Need for Flexible Grazing Management

**Mounir Louhaichi** [1,*], **Mouldi Gamoun** [1], **Farah Ben Salem** [2] **and Azaiez Ouled Belgacem** [3]

1. International Center for Agricultural Research in the Dry Areas (ICARDA), Tunis 1004, Tunisia; M.Gamoun@cgiar.org
2. Institut des Régions Arides, Medenine 4191, Tunisia; farah.bensalem@ira.rnrt.tn
3. Food & Agriculture Organization of the United Nations, Riyadh 11421, Saudi Arabia; Azaiez.OuledBelgacem@fao.org
* Correspondence: M.Louhaichi@cgiar.org; Tel.: +216-71752099

**Abstract:** Resting or grazing exclusion is an effective practice widely adopted to restore degraded, arid rangelands. To understand its effect on plant diversity, we studied Hill's diversity indices during two growing seasons (2017–2019). The experiment consisted of a three-level factorial design with four plant communities subjected to different resting durations (one, two, and three years) compared to continuously grazed areas (control). The results showed that rainfall plays an important role in arid rangeland restoration. Under favorable conditions, one-year grazing exclusion considerably enhanced species richness and evenness diversity compared to longer resting durations under dry to average rainfall conditions. The decision to how long livestock grazing exclusion would last should not be decided upfront as it depends on the climatic and the site-specific conditions. The findings of this study will have vital management implications for development agencies. Knowing that short grazing exclusion with adequate rainfall amount and distribution could be enough and offers a cost-effective technical option to ensure the sustainable restoration of arid rangeland. This flexible grazing management would also be more acceptable by the pastoral communities. Longer resting periods could have detrimental effects on arid rangeland vegetation, in addition to adding more pressure on the remaining rangeland areas open to grazing.

**Keywords:** Hill's diversity; Tunisia; plant community; climate variability; grazing strategy; opportunistic grazing

## 1. Introduction

Arid rangelands occupy approximately 60% of the global rangelands and 70% of drylands and sustain 14% of the world's population and 50% of global livestock [1–3]. The vegetation dynamics of arid rangelands have long been a focus for numerous scientists trying to understand their relationship with climate and human activities [4,5]. Several researchers have challenged the findings of many previous studies that indicate the importance of natural arid rangeland diversity as the major driving force of rangeland health [6–8].

Natural rangelands in Tunisia cover about 33% (5.5 million ha) of the country's total territory, 87% of which are located in the south described as arid and desert areas (45% and 42%, respectively). Southern Tunisia is the driest area of the country as it includes the Great Eastern Erg. These arid rangelands suffer from accelerated degradation due to the combined effects of human pressure and climatic precarity. As human and livestock populations continue to increase, the pressures from livestock grazing due to reduced herd's mobility, encroachment of cultivation into best rangeland sites, and other types of mismanagement—combined with recurrent drought—can ignite local conflicts over resources [9–12]. In fact, the impact of these factors is considered as the most detrimental for rangeland vegetation and reflected through the apparition of invasive species and loss of

key desirable species [12,13]. Given this alarming situation, it is therefore necessary for local government authorities and development agencies to take urgent restoration measures.

Livestock grazing exclusion or the rangeland resting technique is a common traditional practice in the region. In fact, most rangeland improvement projects in southern Tunisia are using the resting technique (locally known as *Gdel* or *Hima*) to restore degraded arid rangelands [14]. This technique is considered by many authors among the most cost-effective restoration practices [7,14,15]. Under the *Hima* system, the grazing land is protected by the local pastoral communities from livestock grazing and wood harvesting for a certain period of time. Unfortunately, certain development agencies who adopted such practice use a fixed resting period of three consecutive years without evaluating the effectiveness of the resting duration on the rangeland ecosystem [16]. In the literature, there are conflicting results. While certain studies conducted in arid environments suggest that species composition and diversity increase with short-term protection, others suggest that long-term benefits are reduced [10,17,18].

Therefore, the objective of this study is to assess the effects of resting duration versus continuous grazing on plant diversity and species richness in the arid rangelands of Tunisia. More specifically, the study seeks to address one main question: how does resting affect plant community structure (e.g., species richness and diversity) across the dominant plant communities (spatial scale), resting duration (temporal scale), and climatic variability (2018 and 2019)?

## 2. Materials and Methods

### 2.1. Site Description

Southern Tunisia has an arid Mediterranean climate with a mean long-term annual rainfall of 80 mm concentrated in autumn–spring (the growing season is September–April) and a dry summer lasting about 4 months during May–August [19]. In Tataouine, the study area (Figure 1), rainfall was 253 mm during the 2017/2018 season and 120 mm during the 2018/2019 season. The highest amount of rain occurred in a short period and was highly variable throughout the year and largely limited to November–April (Figure 2). The soil of the study area is dominated by limestone called Regosol, sandy soil, and sand accumulation. In Tataouine, natural rangelands cover about 1.5 million ha grazed by more than 1.3 million head of sheep and goats (18% of the total livestock population in Tunisia) and 12,000 head of camels (25% of the total livestock population in Tunisia) [20,21].

### 2.2. Experimental Design and Field Measurements

This study targeted the main four plant communities in southern Tunisia dominated by the following species: *Anthyllis henoniana* Coss., *Haloxylon schmittianum* Pomel., *Stipagrostis pungens* Desf., and *Retama raetam* (Forssk.) Webb and Berthel, respectively. Each plant community is characterized by its respective soil type [22]:

o *Anthyllis henoniana* community: growing on limestone soil (Regosol) with high calcium carbonate content, high pH, and low organic matter content.
o *Haloxylon schmittianum* community: it colonizes shallow to deep sands or silty soils and is also found on gravel plains characterized by deep, loose soil with low moisture content, very low organic matter (0.8%), and low soil salinity.
o *Stipagrostis pungens* community: it is located on sandy soil classified as sierozem dominated by mobile sand dunes and characterized by very high-water infiltration and low water retention. The organic matter content is low (less than 1%), pH exceeds 7, and calcium carbonate content is about 10–20%. Sand grains vary from coarse to medium size and are rarely fine.
o *Retama raetam* community: it is found mainly on wadi beds dominated by sand accumulation. This soil is isohumic (subdesertic called sierozem) with a low water retention capacity (range 6–9%) and contains little organic matter (0.5–0.7%). Excessive amounts of calcium carbonate are present, and pH is approximately 8 down the whole profile.

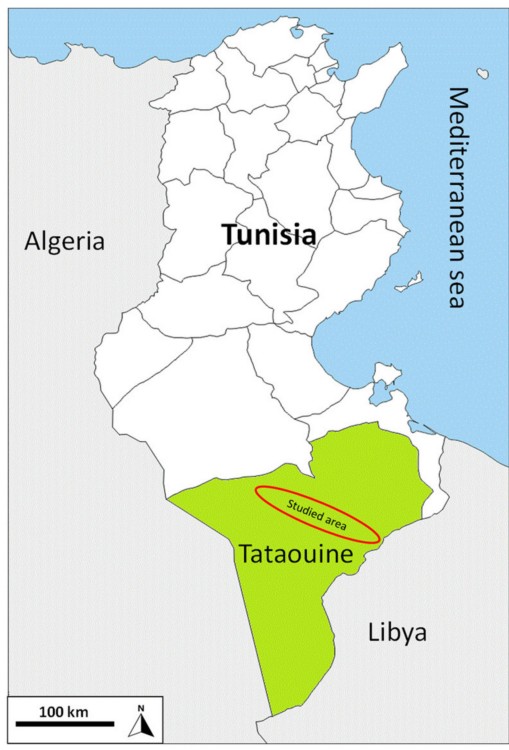

**Figure 1.** Map showing target area in Tataouine, southern Tunisia.

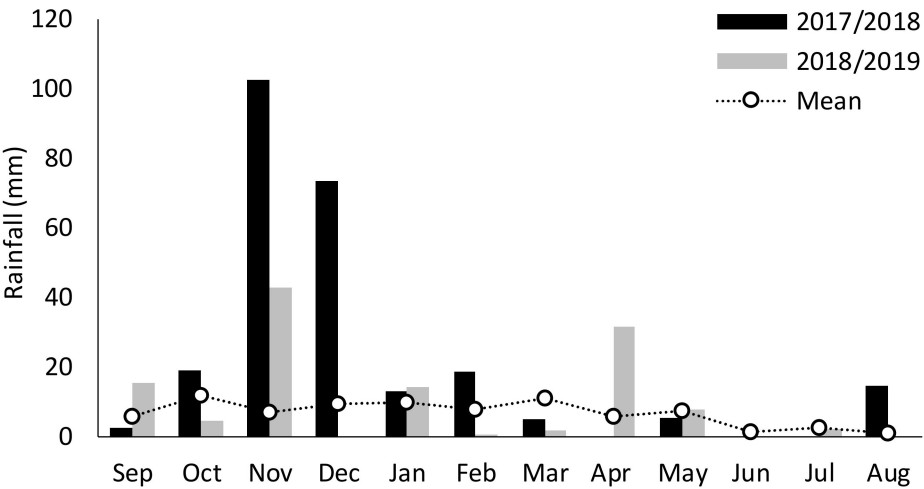

**Figure 2.** Average monthly rainfall for 20 years and during the two experimental seasons 2017/2018 and 2018/2019 in Tataouine, southern Tunisia.

Plant diversity is a critical component in arid rangelands and represents a driving force for rangeland productivity, livestock production, and the well-being of pastoral communities [10,23]. Within this context, the four above targeted rangeland plant communities were selected to examine their plant diversity and species richness in relation to the duration of resting and rainfall variation during two consecutive springs of 2018 and 2019. For each plant community, three different resting periods were tested: 1, 2, and 3 years in 2018. The age categories of resting duration in 2019 become as follows; 2-years resting and 3-years resting, while the site that has been protected for 3 years will be opened for grazing and it is therefore not included in the measurement (Table 1).

**Table 1.** The 22 experimental treatments and their codes applied during the 2017/2018 and 2018/2019 seasons in the four targeted plant communities of southern Tunisia.

| Community | Area (ha) | 2017/2018 | Code | 2018/2019 | Code |
|---|---|---|---|---|---|
| *A. henoniana* | 120 | 1-year rest | 1 | 2 years rest | 14 |
| | 170 | 2 years rest | 2 | 3 years rest | 15 |
| | 20 | 3 years rest | 3 | Open to grazing | |
| *H. schmittianum* | 30 | 1-year rest | 4 | 2 years rest | 16 |
| | 170 | 2years rest | 5 | 3 years rest | 17 |
| | 20 | 3 years rest | 6 | Open to grazing | |
| *S. pungens* | 100 | 1-year rest | 7 | 2 years rest | 18 |
| | 170 | 2 years rest | 8 | 3 years rest | 19 |
| | 20 | 3 years rest | 9 | Open to grazing | |
| *R. raetam* | 30 | 1-year rest | 10 | 2 years rest | 20 |
| | 140 | 2 years rest | 11 | 3 years rest | 21 |
| | 20 | 3 years rest | 12 | Open to grazing | |
| Control | | Continuously grazed | 13 | Continuously grazed | 22 |

Within each plant community and for each treatment, three permanent sampling lines of 50 m were established for vegetation measurements as described by Daget and Poissonet [24]. A fine pin was dropped straight down into the ground at 50-cm intervals along the measuring tape. At each of the 100 points on the line, the contacted plant species were recorded. The proportion of each species recorded in the plant community is equal to the number of points divided by 100. This proportion is frequently used to calculate the Shannon–Wiener diversity index (H′) and Hill's diversity indices. Hill's unified diversity indices $N_0$, $N_1$, and $N_2$ [25] were calculated:

$N_0 = S$ = total number of species present in a sample

$N_1$ is the number of abundant species

$$N_1 = e^{-\sum_{i=1}^{s}\left(\frac{ni}{N} Log2 \frac{ni}{N}\right)}$$

$$N_2 = \frac{1}{\lambda} = \frac{1}{\sum_{i=1}^{s} pi^2}$$

The relative proportion of dominant species can be measured by evenness (or regularity, $E_{20}$), which is estimated as the ratio of very abundant taxa relative to the total number of taxa in a sample using $E_{20} = N_2/N_0$ [26–30]. The $E_{20}$ (Simpson evenness or Hill's ratio) varies between 0 (one species largely dominates all others) and 1 (all species have the same frequency) and is not correlated with species richness:

$$E_{20} = \frac{N_2}{N_0} = \frac{1/\lambda}{N_0} = \frac{1/\sum_{i=1}^{s} pi^2}{N_0}$$

*2.3. Statistical Analysis*

An ANOVA test was applied to study the effects of year (2017/2018–2018/2019), resting durations (1, 2, and 3 years and the control (continuously grazed)) and plant communities (*A. henoniana, H. schmittianum, S. pungens, R. raetam*, and control) on diversity indices $N_0$, $N_1$, and $N_2$, and Evenness Index (Hill's ratio) ($E_{20}$). The analysis was fully factorial. In the current study, there were no true replicates because all sampling plots within each plant community were placed in the same location. Thus, the observations recorded from transects and the quadrats on each site were considered as replicates for the among sites comparison, although they were pseudo-replications of the comparison between the two years as well as the resting durations [31].

Prior to statistical analyses, data were subjected to Shapiro–Wilk W-test to assess the normality assumption. Differences among means were separated using Fisher's Protected

Least Significant Difference when at the 5% significance level Regression analysis and Univariate correlation were used to establish relationships between different variables. All analyses were performed using IBM SPSS-Statistics version 26 (SPSS/IBM, Chicago, IL, USA).

### 3. Results

#### 3.1. Total Number of Species ($N_0$)

The analysis of variance of the total number of species ($N_0$) indicated significant interaction among years, resting durations, and plant communities (Table 2). The *R. raetam* community protected for two and three years recorded the highest number of species ($p < 0.001$) ($N_0$) (51 and 48, respectively) during the first year (2018). The lowest species number was within the *A. henoniana* community protected for three years in 2018 (17 species) followed by the control in 2019 with 18 species (Figure 3). Among the plant communities, the highest number of species was in the *R. raetam* community followed by *H. schmittianum*, while the lowest value of $N_0$ was recorded in the control. The significant effect of resting duration on No is more visible in the first year but decreased with longer resting duration mainly in 2019.

**Table 2.** ANOVA table for three factorial experimental design to assess the effects of resting duration versus continuous grazing on plant diversity and species richness in arid rangelands of Tunisia.

| Source | df Effect | Total Number of Species ($N_0$) | | Number of Abundant Species ($N_1$) | | Number of Very Abundant Species ($N_2$) | | Evenness Index ($E_{20}$) | |
|---|---|---|---|---|---|---|---|---|---|
| | | F | p | F | p | F | p | F | p |
| Plant community (A) | 3 | 47.47 | <0.0001 | 6.28 | 0.0009 | 9.65 | <0.0001 | 13.58 | <0.0001 |
| Resting durations (B) | 2 | 13.95 | <0.0001 | 3.18 | 0.0488 | 6.26 | 0.0034 | 1.68 | 0.1946 |
| Year (C) | 1 | 42.58 | <0.0001 | 25.56 | <0.0001 | 16.45 | 0.0001 | 1.49 | 0.227 |
| A × B | 6 | 14.68 | <0.0001 | 3.5 | 0.4641 | 2.65 | 0.024 | 1.12 | 0.364 |
| A × C | 3 | 27.01 | <0.0001 | 0.87 | 0.0049 | 0.48 | 0.695 | 0.65 | 0.5867 |
| B × C | 1 | 26.63 | <0.0001 | 0.18 | 0.6755 | 0.6 | 0.4427 | 1.3 | 0.2589 |
| A × B × C | 3 | 0.35 | <0.0001 | 1.81 | 0.1546 | 1.81 | 0.1554 | 1.11 | 0.3532 |

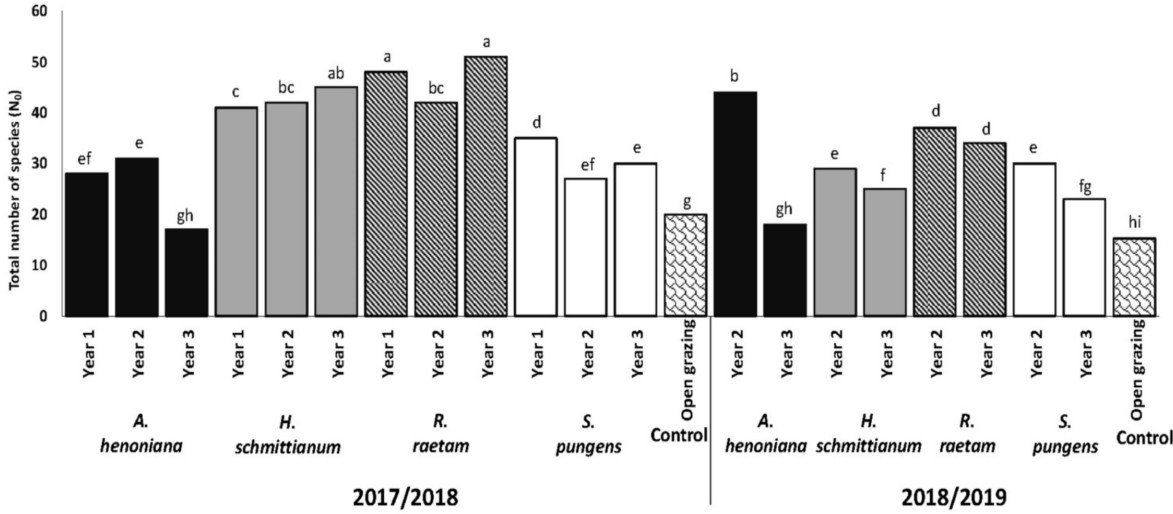

**Figure 3.** Total Number of Species ($N_0$) recorded in four plant communities (*A. henoniana*, *H. schmittianum*, *S. pungens*, *R. raetam*) and control sites managed under different resting durations (1, 2, and 3 years and the control) during 2018 and 2019, Southern Tunisia. The set of bars with different small letters denote that the difference is significant at $p < 0.05$.

The 2018 year had a beneficial impact on $N_0$. Relatively higher numbers of species were observed in all plant communities including the control sites during the rainy year of 2018 compared to the normal year 2019 (Figure 3).

### 3.2. Number of Abundant Species ($N_1$)

Plant community, resting duration, and year had significant effects on number of abundant species ($p < 0.05$). Interaction between year and plant community factors also significantly influences number of abundant species.

Diversity of abundant species ($N_1$) decreased gradually with the increase of resting duration. The highest value was recorded at the 1-year rest duration (25 species) followed by the 2-year rest duration (17 species). The lowest value (13 species) was registered in the control (Figure 4).

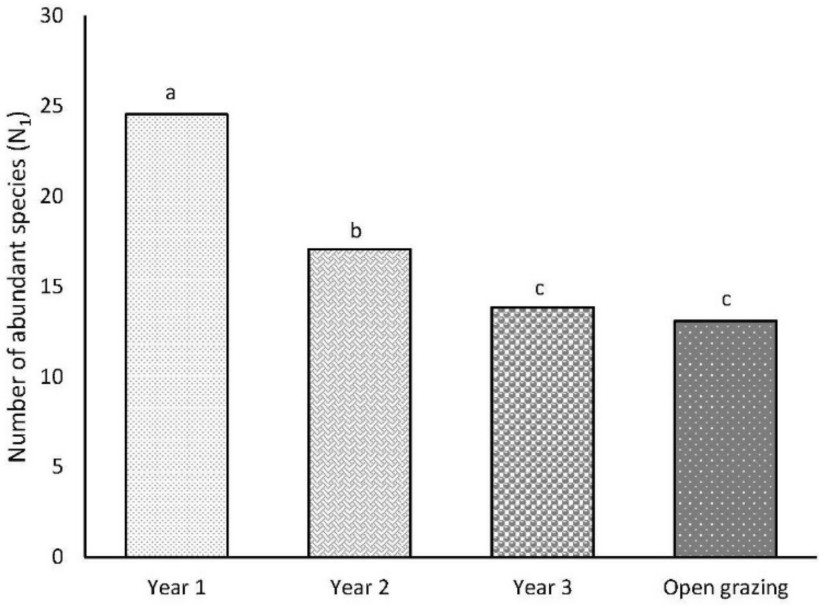

**Figure 4.** Number of abundant species ($N_1$) recorded during four different resting durations (1, 2, and 3 years) compared to the control (open grazing). Values are means of four plant communities (*A. henoniana*, *H. schmittianum*, *S. pungens*, *R. raetam*) and control in two years (2018 and 2019). The set of bars with different small letters denote that the difference is significant at $p < 0.05$.

The ANOVA showed a highly significant effect of the year on the presence of abundant species $N_1$ within all plant communities and the control site ($p < 0.01$). A greater $N_1$ was recorded for all sites in the rainy year (2018). The abundant species values in the *R. raetam*, *A. henoniana*, and *H. schmittianum* communities were the highest in 2018 (26, 23, and 13 species, respectively). The values of abundant species in the grazed sites followed the same trend and recorded 11 more species in 2018 (compared to 15 species in 2019). In contrast, the *S. pungens* community tended to record the lowest diversity of abundant species ($N_1$) in both years (Figure 5).

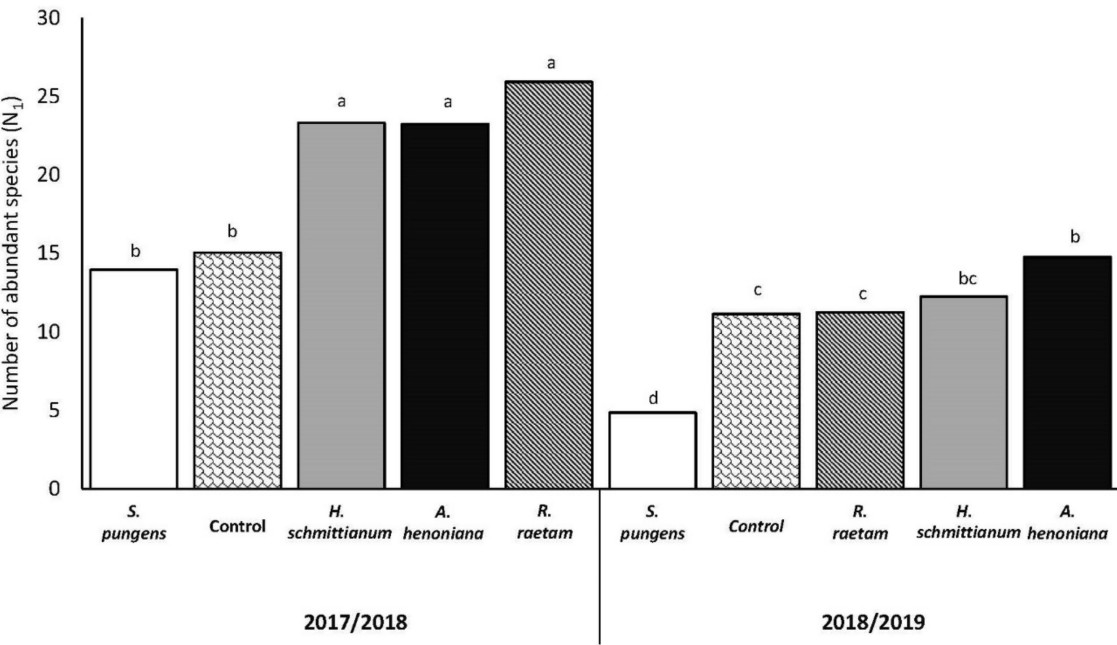

**Figure 5.** Number of abundant species ($N_1$) recorded in four plant communities (*A. henoniana, H. schmittianum, S. pungens, R. raetam*) and control sites during 2018 and 2019 years, Southern Tunisia. Values are means of four different resting durations (1, 2, 3 years and control). The set of bars with different small letters denote that the difference is significant at $p < 0.05$.

### 3.3. Number of Very Abundant Species ($N_2$)

The number of very abundant species ($N_2$) varies significantly across plant community, resting period, year, and plant community and grazing period interaction ($p < 0.05$). A total of 72 very abundant species were recorded in the four plant communities and the control during two years, of which 10 and 9 species occurred in the *A. henoniana* and *R. raetam* plant community sites subjected to first year rest. The lowest number of very abundant species was recorded in *S. pungens* community sites after two- and three-years rest (Figure 6). However, for all plant communities, $N_2$ tended to decrease with the increase of resting duration.

The yearly rainfall condition had a highly significant effect on very abundant species $N_2$ ($p < 0.01$). The mean $N_2$ was higher in 2018 than in 2019 (5.9 vs. 3.9) (Figure 7).

### 3.4. Evenness Index ($E_{20}$)

The plant communities × year, plant communities × resting duration, and plant communities × year × resting duration interactions, and the main effect of year and resting did not affect the evenness index ($E_{20}$). However, $E_{20}$ varied significantly with the plant community including the control sites ($p < 0.001$). The value of $E_{20}$ exceeded 6.5 species in the *A. henoniana* community and decreased by half in the *S. pungens* plant community sites (3.4), recording the lowest value among all plant communities. Similar $E_{20}$ values were registered in the open grazed sites, and *H. schmittianum* and *R. raetam* sites (5 species) (Figure 8).

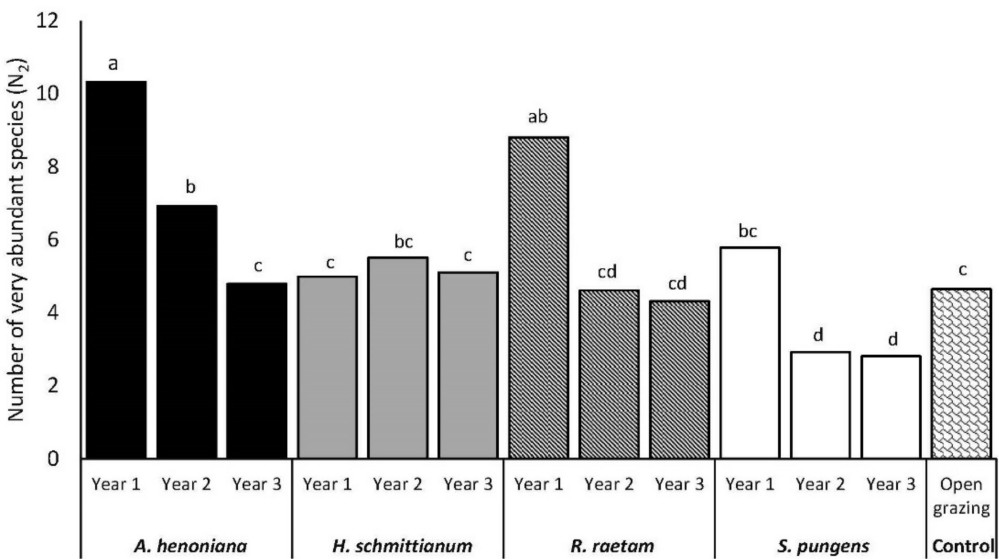

**Figure 6.** Number of very abundant species ($N_2$) recorded in four plant communities (*A. henoniana*, *H. schmittianum*, *S. pungens*, *R. raetam*) under four different resting durations (1, 2, and 3 years and control (open grazing)) in southern Tunisia. Values are means of 2018 and 2019 years. The set of bars with different small letters denote that the difference is significant at $p < 0.05$.

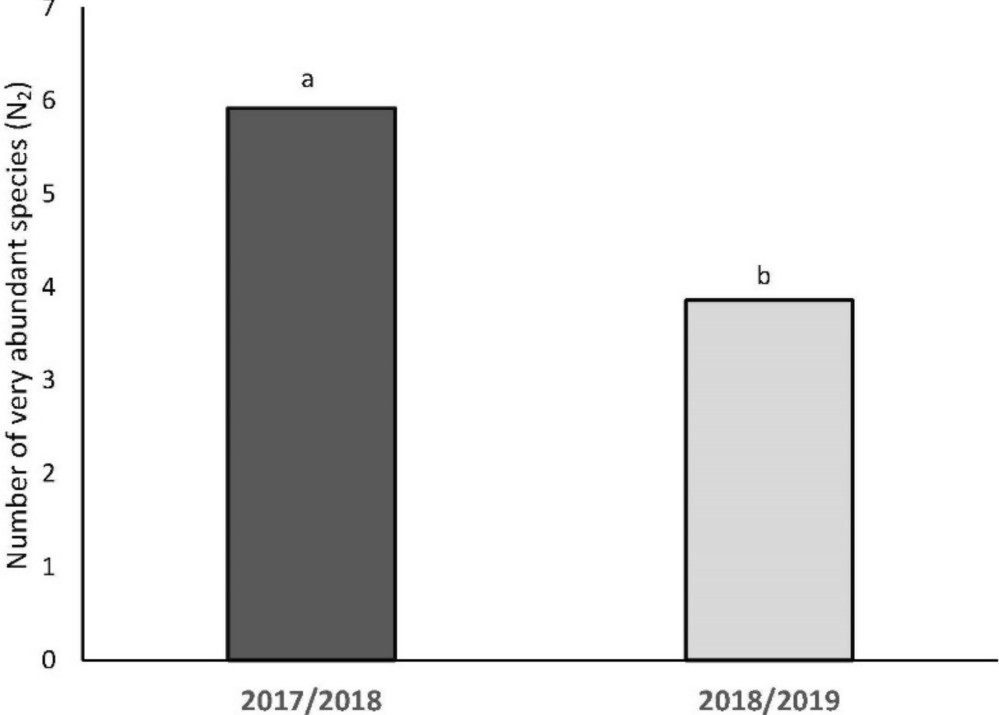

**Figure 7.** Number of very abundant species ($N_2$) recorded during 2018 and 2019 years, Southern Tunisia. Values are means of four plant communities (*A. henoniana*, *H. schmittianum*, *S. pungens*, *R. raetam*) subjected to different resting durations (1, 2, and 3 years and control (open grazing)). The set of bars with different small letters denote that the difference is significant at $p < 0.05$.

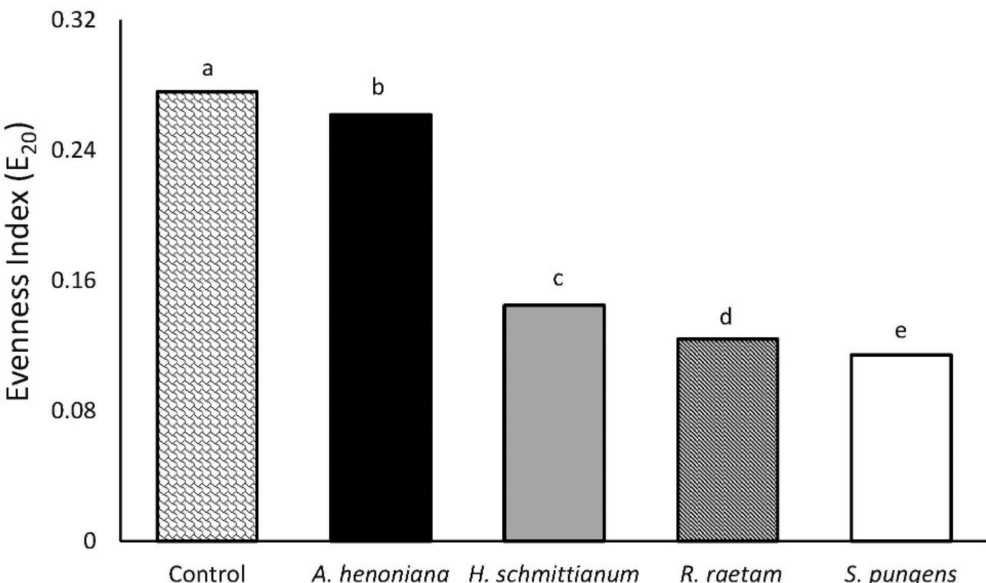

**Figure 8.** Evenness Index ($E_{20}$) recorded in four plant communities (*A. henoniana*, *H. schmittianum*, *S. pungens*, *R. raetam*) and control, Southern Tunisia. Values are means of four different resting duration periods (1, 2, 3 years and control (open grazing)) recorded during the 2018 and 2019 years. The set of bars with different small letters denote that the difference is significant at $p < 0.05$.

Although the effect of resting duration and the experimental year on species evenness index was not significant, the evenness index decreased slightly mainly after one year of resting. The relatively drier year (2019) followed the same trend and had a lower $E_{20}$ value compared to the good rainfall year (2018).

*3.5. Correlations Between Hill's Diversity Indices*

All Pearson correlation coefficients between the studied variables related to species richness and plant diversity were highly significant ($p < 0.01$).

The simple linear regression model showed that the correlations of $N_0$ with $N_1$ and $N_2$ were positive ($r = 0.492$ and $0.318$, respectively, both $p < 0.01$). However, the correlation between $N_0$ and $E_{20}$ was significantly negative ($r = -0.542$, $p < 0.01$). The correlations between $N_1$ and $N_2$ were higher and strongly positive (r = 0.897, $p < 0.01$). Unlike the negative relationship between $E_{20}$ and $N_0$, the correlations of $E_{20}$ with $N_1$ and $N_2$ were positive ($r = 0.326$ and $0.546$, respectively, both $p < 0.01$).

**4. Discussion**

The impact of biotic stress, particularly grazing pressure, on plant diversity is quite controversial [32,33]. On one hand, grazing is considered as a key factor to promote diversity [34,35], on the other hand, grazing can reduce plant diversity and lead to the homogenization of rangeland [36,37]. Furthermore, other studies carried out in arid areas have shown that climate variability is more important in affecting plant diversity than grazing, and moderate grazing does not damage vegetation [10,38].

The assessment of different resting durations in four plant communities during the two different years (favorable and normal) showed distinct variability in Hill's diversity indices, which characterize plant communities of arid rangelands. Our results showed the effect of grazing exclusion compared to free continuous grazing on the spatial and temporal dynamics of rangeland vegetation in this arid area.

Grazing pressure is recognized as a regulator of vegetation dynamics, altering diversity indices such as species richness and evenness [27,39,40]. In our study, these indices varied significantly with resting duration. The $N_0$ was higher when the rangeland site was rested, with an increasing trend of $E_{20}$ compared to treated areas, where the lowest value occurred

for the *S. pungens* community subjected to two years of rest. Both abundant and very abundant species remained stable whatever the management mode. Our findings confirm several previous studies conducted in arid rangelands in which resting improved $N_0$ [41,42]. In contrast, the $E_{20}$ index increased in the continuously grazed sites, corroborating the results of many studies that showed an increase of this index with disturbance [27,43,44]. The decrease in $E_{20}$ after applying the rest technique supports the "competitive exclusion hypothesis" described by Grime [45,46] and Huston [47]. It seems that competition is inversely related to diversity as an increase in the intensity of competition results in a decrease in the evenness and, eventually, the species richness [48,49].

Although arid-zone vegetation is mostly composed of xerophytes, rainfall intensity and distribution are the major drivers of species diversity and can potentially lead to changes in species composition. In this study, the year with good rainfall played a more important role than any management mode in explaining species diversity. All species diversity indices ($N_0$, $N_1$, $N_2$, and $E_{20}$) varied significantly with yearly climatic conditions, with higher values in the rainy year. Several recent studies showed that rainfall is a key driver of rangeland structure and function in arid areas [13,50,51]. However, exceptional highly favorable years can greatly change the diversity of arid rangelands. Additionally, good rainfall in both amount and distribution can increase species diversity by enhancing the establishment and survival of new seedlings. This statement is confirmed by the indigenous knowledge of the local pastoralists and rangeland researchers in southern Tunisia. It is known that when the seasonal rainfall occurring between September and December does not exceed 60 mm, at best we are expecting a normal year with an average rangeland productivity. However, when the rainfall exceeds 80 mm during the same period, this will definitely lead to a favorable year with high rangeland production [52]. Surprisingly, Tielbörger et al. [53] found that species richness did not change after nine years of drought, which they attributed to a 'climatic comfort zone' linked to species adaptation.

In general, in drylands, the increasing demand for water availability following rain events may cause considerable disruption to diversity and species richness [54,55].

Our findings showed an increase in species richness for the *R. raetam* community on sandy soil (Appendix A), but the lowest species richness was recorded for the *A. henoniana* community on limestone soil. However, this response varied with soil and vegetation type. These results corroborate the findings of other studies for arid ecosystems [56,57]. In southern Tunisia, Floret and Pontanier [58] explained that rainfall and soil type were decisive factors and higher response of vegetation and soil moisture levels were observed in deep sandy soils compared to limestone soils. This is opposite to what has been found in other arid areas of the world [59,60].

In our study, $E_{20}$ was negatively correlated with $N_0$, but not with $N_1$ and $N_2$. Among all treatments, the lowest $E_{20}$ value was for the *S. pungens* community subjected to three years of resting. The low $N_1$ and $N_2$ values, and therefore low $E_{20}$, in the *S. pungens* community, despite resting, can be explained by an increased competition for water. It is clear that competition for water resources is important in determining the establishment, survival, and reproduction of annual plants in arid ecosystems. *Stipagrostis pungens* is a psammophyte well known for its role in stabilizing sand dunes, due to its well-developed lateral and superficial root system [61]. During periods of light rainfall, *S. pungens* roots can exploit soil water from surface horizons. The well-developed root system of *S. pungens* and soil-moisture loss in sand dunes due to high rates of infiltration means that small amounts of water are available for annual plants such as *Cutandia dichotoma* and *Schismus barbatus*. Hence, sand dunes colonized by *S. pungens* community have low flora richness in terms of $N_1$ and $N_2$. In contrast, resting increased $E_{20}$ in the *A. henoniana* community. *Anthyllis henoniana* has long taproots, which can penetrate the very compacted soil to reach water at great depths, enhancing the growth of annual plants mainly in the understory of perennial shrubs due to the available soil surface moisture. This positive interaction between species indicates the facilitation by *A. henoniana* for the development of annual vegetation in an arid environment. Understanding the existence of facilitation, also called mutualism,

between plants is considered a critical process for the development of effective tools for achieving ecosystem restoration objectives [62]. Similarly, overgrazing and trampling leads to increasing evenness that may be explained by greater adaptation of species and therefore promoting complementarity and facilitation [63]. Continuous grazing of arid vegetation decreases species richness and therefore only a few very abundant species can persist with similar distributions. These species, which are resistant to environmental disturbance, become more vulnerable to continuous grazing and are consequently rare and threatened by extinction. We predict that evenness is proportional to the effect of disturbance and increases with grazing pressure in arid rangelands. This will seriously affect the resilience of rangeland species.

Furthermore, this study investigated the effects of rest periods in relation to rainfall regime and showed a significant interaction of resting technique with the amount and distribution of rainfall. $N_0$ was higher in the *R. raetam* community with three years of rest. The highest $N_1$ was in the *R. raetam* community subjected to a one-year rest followed by *A. henoniana* and *H. shmittianum* communities rested for two years. However, the highest $N_2$ values were registered in the *A. henoniana* and *R. raetam* communities rested for one year, followed by the *A. henoniana*, *H. shmittianum*, and *R. raetam* communities rested for two years. The decrease in annual rainfall during the second year of the experiment caused lower $N_0$, $N_1$, and $N_2$ despite the increased rest period. It is likely that within the *A. henoniana* community there was greater mutualism and positive interactions between species regardless of rainfall conditions and the rest duration. $E_{20}$ was significantly greater at high rainfall compared to all other treatments. The high $E_{20}$ of the *A. henoniana* community protected for one year indicates the facilitation effects that *A. henoniana* provided to annual plants to complete their growth.

Regardless of climate condition and plant community, all diversity indices ($N_0$, $N_1$, $N_2$, and $E_{20}$) were higher at the sites subjected to only one year of rest than those that were protected for longer periods or that were continuously grazed. The results also indicated that, under reasonable rainfall amounts and distributions, resting a previously grazed rangeland for one year was sufficient to maintain $N_0$. Consequently, our results support other findings suggesting that excluding grazing is one of several strategies that need to be adopted to facilitate and restore rangeland biodiversity [64,65].

In general, annual plants are less resistant to trampling, grazing, and drought than perennial plants [66,67]. Complete protection from grazing can be beneficial to productivity and diversity [37,68]. The short-term effect of grazing exclusion during a favorable year was found to be related to soil surface compaction caused by grazing [68]. One beneficial effect of trampling is the creation of a patchwork that results in a microclimatic change, with positive effects on seedling establishment due to collecting runoff in the soil surface depressions. In arid rangelands, grazing exclusion at specific periods is required. Nonetheless, the duration of these rest periods depends on the plant community, the degree of prior grazing pressure, and the climatic conditions [14].

As the pressure on rangelands continues to mount, it would be wise to combine pastoral indigenous knowledge with science-based evidence to tackle big challenges and find solutions that are technically sound and socially accepted. In southern Tunisia, even though the resting technique requires total exclusion of animal grazing, the pastoral communities have always respected the rules based on social awareness. The approach has been successfully implemented through closely involving tribal institutions, with one of the main objectives to strengthen sustainable rangeland restoration through the revival of the traditional grazing system known as *Gdel* [14]. However, there was no consensus to how long the resting period should be. Results from this study show that there are no valid justifications for banning livestock grazing for three consecutive years. In fact, as demonstrated, short-term exclusion (1 year), with sufficient precipitation greatly increased plant diversity underlining the high resilience capacity of these arid ecosystems. These findings could be very useful for policy-makers in reviewing the current rangeland

management strategy in the country to offer more flexibility for the pastoral communities to sustainably manage their natural resources.

## 5. Conclusions

In arid rangelands, plant diversity is one of the most important key functions of healthy ecosystems. Human-induced disturbances combined with recurrent droughts represent the main causes of ecosystem disequilibrium leading to threats to key plant species and consequently its floristic cortege. Inappropriate grazing that causes declines in plant diversity alerts rangeland managers to consider suitable management changes. During the 2017/2018 period, the recoded rainfall in the southern arid rangelands of Tunisia was 200% higher than the long-term average, which had a considerable positive impact on species richness, abundance, and evenness. Our findings suggest that for the arid rangelands of southern Tunisia, the exceptional favorable wet season was very helpful in mitigating the negative effects of continuous grazing pressure and recurrent drought regardless of the duration of the resting period. This scenario led to significant increases in plant diversity for most plant communities, mainly *A. henoniana* and *R. raetam*. Maintaining rangeland plant diversity serves as an insurance policy for the survival of healthy rangelands to provide a sustainable ecosystem of goods and services. Under these conditions, it is important to consider a flexible approach to grazing management, depending on climate conditions and site specificity.

This study presented an opportunity to observe the short-term changes in arid rangeland plant diversity with respect to resting duration and climate variability. Its findings are expected to contribute to the development of sustainable rangeland management strategies across the dry areas to strengthen the resilience of the pastoral communities.

**Author Contributions:** Conceptualization, M.L., M.G., A.O.B., and F.B.S.; methodology and data collection, M.L., M.G., A.O.B., and F.B.S.; data analysis, M.G., and M.L.; writing—original draft, M.G.; writing—review and editing, M.L. and A.O.B.; project administration, M.L.; funding acquisition, M.L. All authors have read and agreed to the published version of the manuscript.

**Funding:** Please add: This research was funded by CGIAR Research Program on Livestock Agri-Food Systems., grant number 200081.

**Data Availability Statement:** Not applicable.

**Acknowledgments:** This work was supported by the International Center for Agricultural Research in the Dry Areas (ICARDA), the Institute of Arid Lands (IRA) in Tunisia, the Office of Livestock and Pastures (OEP), and the CGIAR Research Program on Livestock (CRP Livestock). The authors thank Fethi Gouhis, E. Belfekih and M. Abdelkader (OEP) and Said Debbabi and Abdennacer Rached (IRA) for their support during the field data collection.

**Conflicts of Interest:** The authors declare no conflict of interest.

## Appendix A

List of encountered plant taxa in studied plant communities and their abundance based on their proportion in the community: - absent, + rare, ++ abundant, +++ very abundant.

| Species | *A. henoniana* | *H. schmittianum* | *S. pungens* | *R. raetam* | Control |
|---|---|---|---|---|---|
| *Aegilops ventricosa* | + | - | - | - | - |
| *Anacyclus clavatus* | +++ | +++ | - | ++ | + |
| *Anacyclus monanthos* | ++ | ++ | - | + | - |
| *Anthyllis henoniana* | +++ | + | + | + | ++ |
| *Argyrolobium uniflorum* | - | ++ | + | + | ++ |
| *Arnebia decumbens* | - | - | - | + | - |
| *Artemisia campestris* | - | - | + | - | - |
| *Artemisia herba-alba* | + | - | - | - | - |
| *Asphodelus tenuifolius* | - | ++ | ++ | ++ | - |
| *Astragalus caprinus* | - | - | + | - | - |
| *Astragalus corrugatus* | ++ | + | ++ | + | - |

| Species | A. henoniana | H. schmittianum | S. pungens | R. raetam | Control |
|---|---|---|---|---|---|
| Atractylis cancellata | + | - | - | - | - |
| Atractylis carduus | - | + | + | + | - |
| Atractylis serratuloides | +++ | - | - | - | - |
| Avena sterilis | + | - | - | - | - |
| Bassia muricata | - | - | - | + | - |
| Bromus rubens | + | - | - | - | - |
| Calendula arvensis | + | + | + | + | - |
| Centaurea furfuracea | - | ++ | - | + | - |
| Cleome amblyocarpa | + | + | - | + | - |
| Convolvulus supinus | - | + | - | + | - |
| Cuscuta epithymum | + | - | - | - | - |
| Cutandia dichotoma | - | +++ | +++ | +++ | - |
| Cynara cardunculus | + | - | - | - | - |
| Cynodon dactylon | - | - | - | +++ | - |
| Daucus sahariensis | - | ++ | +++ | ++ | + |
| Deverra denudata | + | - | - | - | - |
| Deverra tortuosa | - | - | - | + | - |
| Diplotaxis harra | + | + | - | - | - |
| Diplotaxis simpelx | - | - | + | - | - |
| Echium humile | ++ | - | ++ | - | - |
| Enarthrocarpus clavatus | + | - | - | - | - |
| Erodium arborescens | ++ | - | - | - | - |
| Erodium crassifolium | + | - | - | - | - |
| Erodium laciniatum | + | ++ | + | + | - |
| Erucaria pinnata | ++ | - | - | - | - |
| Euphorbia retusa | - | + | - | - | - |
| Euphorbia terracina | + | - | - | - | - |
| Fagonia cretica | - | + | - | + | - |
| Fagonia glutinosa | - | ++ | + | + | ++ |
| Filago germanica | - | ++ | - | ++ | +++ |
| Gymnocarpos decander | +++ | + | + | + | ++ |
| Haloxylon schmittianum | - | +++ | + | ++ | ++ |
| Haloxylon scoparium | - | - | - | - | +++ |
| Hedysarum spinosissimum | + | + | ++ | + | - |
| Helianthemum kahiricum | +++ | ++ | - | + | +++ |
| Helianthemum nummularium | + | - | - | - | - |
| Helianthemum sessiliflorum | + | +++ | ++ | + | - |
| Hippocrepis areolata | - | - | ++ | + | - |
| Hordeum marinum | + | - | - | - | - |
| Ifloga spicata | - | +++ | - | ++ | +++ |
| Kickxia aegyptiaca | ++ | ++ | - | + | - |
| Koelpinia linearis | - | + | ++ | + | - |
| Launaea angustifolia | - | - | ++ | - | - |
| Launaea fragilis | +++ | ++ | +++ | ++ | ++ |
| Launaea glomerata | - | + | - | + | - |
| Launaea nudicaulis | - | - | - | + | - |
| Limonium pruinosum | - | - | - | - | +++ |
| Linaria laxiflora | - | + | - | - | - |
| Lotus halophilus | - | +++ | ++ | +++ | - |
| Lygeum spartum | + | - | - | - | - |
| Matthiola longipetala | - | - | ++ | ++ | ++ |
| Medicago minima | - | ++ | +++ | ++ | + |
| Neurada procumbens | - | ++ | - | ++ | - |
| Nonea calycina | + | - | - | - | - |
| Pallenis hierochuntica | - | ++ | - | ++ | - |
| Paronychia arabica | - | - | ++ | ++ | + |
| Picris asplenioides | ++ | - | - | - | - |
| Plantago albicans | ++ | +++ | +++ | ++ | ++ |
| Plantago coronopus | - | + | - | + | - |
| Plantago ovata | - | ++ | ++ | + | + |
| Polygonum equisetiforme | + | - | - | - | - |
| Reaumuria vermiculata | ++ | - | - | - | - |
| Reichardia tingitana | ++ | - | - | - | - |
| Reseda alba | - | - | + | - | - |
| Retama raetam | - | - | ++ | +++ | - |
| Rhanterium suaveolens | - | - | + | - | - |
| Salsola vermiculata | + | - | + | - | - |
| Salvia aegyptiaca | - | + | - | + | - |
| Savignya parviflora | - | ++ | - | - | ++ |
| Schismus barbatus | ++ | + | ++ | +++ | ++ |
| Scorzonera undulata | + | + | - | + | - |
| Senecio glaucus | + | + | - | + | - |
| Silene villosa | - | + | ++ | ++ | - |
| Stipa capensis | ++ | ++ | - | - | - |
| Stipa lagascae | - | - | - | + | - |
| Stipa tenacissima | - | - | - | + | - |
| Stipagrostis pungens | - | - | +++ | - | - |
| Thesium humile | + | + | - | + | - |

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
