# Peer review of "Rangeland Biodiversity and Climate Variability: Supporting the Need for Flexible Grazing Management"

_sustainability, doi:10.3390/su13137124_

Round 1

Reviewer 1 Report

The paper “Climate change effects on rangeland biodiversity: supporting the need for flexible grazing management” by Mounir Louhaichi et al. is an interesting paper on the restoring of degraded rangelands. However, it needs some changes.

Introduction: -

Lines 29-30, I do not have a specific knowledge of the topic, but I think it is an exaggeration that 1 billion people live in these areas and with 50% of the livestock. Please to check;

Lines 39-40, can you please to better explain the reasons of human pressure which is cause of overgrazing? Are they perhaps due to the rules on land ownership, local traditions and customs? Or simply the need to feed the population?
However, there are no attempts to improve productivity with suitable food supplements, with a reduction in the number of animals, with shifts in grazing, etc.? Is anyhow a nomadic system?

Materials and Methods: -

Line 85, The This study…? Furthermore, please to better explain the meaning of “private rangelands”, are they fenced? Are the four areas far away each other and with different owner?

Line 93, is correct a so low value of organic matter?

Lines 106-107, please to explain in which season (the same for the 4 areas?) the measurements occurred and why the weight of the plants were not included. Moreover, every plant has been considered or only those eaten by livestock?

Table 1, control (mosaic vegetation) what does it means? That still grazing areas of the 4 types were evaluated and average values were used for comparison (please to explain, also the number of sites which has been evaluated);

Lines 125-150, because the pastures are important as animal feed supplier, it would be useful to have a better explanation of the 4 indices meaning in relation to animal requirements (and whether possible to make some quantitative estimation)

Results and discussion: -

It is for me difficult to suggest specific changes but looking to your lines 410-420 were the consequences on yield and livestock performance are been showed, as well as the shepherd traditions, I suggest modifying the results and much more the discussion to make easier the comprehension of No, N1, N2 and E20 meaning (as previously suggested in Materials and Methods).

Conclusions

Line 422, This In arid…?

In general, I agree with the author’s suggestions, but the role of rainfall would be attenuated because it cannot be modified.

Author Response

Review Report Form 1

Introduction:

Comment

Lines 29-30, I do not have a specific knowledge of the topic, but I think it is an exaggeration that 1 billion people live in these areas and with 50% of the livestock. Please to check;

Response

The information has been checked: Dryland support 14% of the world’s inhabitants and dryland rangelands support approximately 50% of the world’s livestock

In fact, a publication by the United Nations Environment Management Group titled “Global Drylands: A UN System wide response” they quote “More than two billion people depend on the world’s arid and semi-arid lands. https://bit.ly/2OJRDMn . Furthermore, you may kindly check the United Nation Decade website where it was stated  that the dry lands are the home of The total drylands 2.1 billion along with 50% of the global livestock (meaning they are the home to one in three people in the world today). https://www.un.org/en/events/desertification_decade/whynow.shtml

Comment

Lines 39-40, can you please to better explain the reasons of human pressure which is cause of overgrazing? Are they perhaps due to the rules on land ownership, local traditions and customs? Or simply the need to feed the population?

Response

More information about the reasons for overgrazing is added
However, there are no attempts to improve productivity with suitable food supplements, with a reduction in the number of animals, with shifts in grazing, etc.? Is anyhow a nomadic system?

Response

Unfortunately, government are not tackling the underlaying causes of rangeland degradation and for political reasons they cannot force reduction in animal number. Furthermore, the traditional grazing system (nomadic) has been dismantled as pastoral communities are forced to settle near villages and water points.

Materials and Methods:

Comment

Line 85, The This study…? Furthermore, please to better explain the meaning of “private rangelands”, are they fenced? Are the four areas far away each other and with different owner?

Response

All the text has been revised and the term of “private rangelands” is deleted

Comment

Line 93, is correct a so low value of organic matter?

Response

It has been corrected

Comment

Lines 106-107, please to explain in which season (the same for the 4 areas?) the measurements occurred and why the weight of the plants were not included. Moreover, every plant has been considered or only those eaten by livestock?

Response

Rangeland inventorying is conducted during the peak standing crop which coincides with the spring season (from March to April depending on weather conditions. In our case the measurements were carried out in March 2018 and 2019 for the four plant communities. The main objective of our study is to investigate their plant diversity and species richness, so we don't need to measure plant biomass.

Comment

Table 1, control (mosaic vegetation) what does it means? That still grazing areas of the 4 types were evaluated and average values were used for comparison (please to explain, also the number of sites which has been evaluated);

Response

Because the control is open to grazing, the dominant desirables and palatable species have disappeared. The site is dominated by heterogeneous unpalatable and herbaceous species.

Comment

Lines 125-150, because the pastures are important as animal feed supplier, it would be useful to have a better explanation of the 4 indices meaning in relation to animal requirements (and whether possible to make some quantitative estimation)

Response

The 4 indices are related mainly to species richness which give us an idea about the available feed resources to animals. The abundance of all available species based on their proportion in the plant community and their palatability as indicated in appendix A that give the required information related to the contribution of each species to animal feeding.

Results and discussion:

Comment

on yield and livestock performance are been showed, as well as the shepherd traditions, I suggest modifying the results and much more the discussion to make easier the comprehension of No, N1, N2 and E20 meaning (as previously suggested in Materials and Methods).

Response

The Materials and Method, Results and Discussion sections have been revised taking into consideration all the comments suggested by the reviewer.

Conclusions

Comment

Line 422, This In arid…?

Response

It has been corrected

Reviewer 2 Report

I have reviewed the manuscript titled “Climate change effects on rangeland biodiversity: supporting the need for flexible grazing management”. It describes the work carried out in arid rangeland conditions on grazing exclusion and pasture recovery. It is a relevant topic for these biomes and adds information to the field. However, the design has some critical flaws that should be addressed before it can be considered for publication.

General comment: the main flaw of the manuscript is the statistical approach and its description. The authors state that a one-way ANOVA was used to analyse the difference between main factors (grazing exclusion, rest duration, vegetation type and year effect). However, when presenting the results, they mention “interaction effects” – these effects can only be tested with various types pf factorial designs. Given that there are many factors being considered and several potential comparisons, the authors should seek for statistical advice on the appropriate experimental design and statistical model, and on the analysis to be performed (e.g. contrasts). Once the statistical approach is proved correct, the authors should amend the Results and Discussion section accordingly. Do not use “significant” or significantly” – that is redundant. You should not describe differences between groups if they were not statistically significant. Please present the actual p value (instead of “<0.05”). Please add lowercase letter to figures with more than two treatments to separate means. Do not use the term “parameter” interchangeably with “variable”. You are measuring variables (sample mean) to try to estimate parameters (population mean).

Specific comments:

Title: weather variation between two consecutive years is not climate change. The title should be amended to describe the actual work presented.

Introduction

L43-56: please rephrased the sentence since it is not well structured.

“Continuous grazing” – the correct term is “continuous stocking” as per Allen et al. (2011) (doi: 10.1111/j.1365-2494.2010.00780.x). Please find a replace the term throughout the manuscript.

Material and Methods

L76: “more than 1,349,410 head” – this looks like a very precise number to be referred as “more than”. It could be referred as “more than 1.3 million heads” to match the style for the area covered (“about 1.5 million ha”).

Figure 1 – what is the period for the “mean” series? Usually a climatic means is taken from 30 years of data. Also in this figure “annual and monthly” -  I can see only the monthly values. Please amend.

L85: “This study…” (delete “The”). Should we assume that each private rangeland contained all the four vegetation types? Or should we assume that each private rangeland had one only vegetation type? This is not clear at all. Please clarify it.

L106: please state what are the spring months and how the plant communities were selected.

L118: what do the authors mean by “minimal area”? Please clarify it in the manuscript.

L123: “is equal to the number of points multiplied by 100” -  I would have thought a proportion is a number resulting from counting unit DEVIDED into 100. Please clarify or amend.

L125-128: these lines should be removed from this section since they are discussing the methodology. They could be moved to the Discussion section if they are relevant to discuss any of the findings.

L128-129: this may be rephrased as “Hill’s unified diversity indices N0, N1 and N2 [31] were calculated:”

Equations: only one equation for N1 should be presented – the authors have presented a kind of demonstration about why N1 equals H’. Please only use the one that uses p (in the second line of N1 equations shown) to match the style of the equation for N2.

L139-146: these lines should be removed from this section since they are discussing the methodology. They could be moved to the Discussion section if they are relevant to discuss any of the findings. Please start this paragraph with “The relative proportion of dominant species…”

Statistical analysis

How was the “private rangeland” considered into the model? Was is a block? Was is the “main-plot” (assuming a potential split-plot design)? Was it just another factor (in turn associated with the vegetation type)?

How was year included in the model? Just as a main factor (“year effect”) or as a blocking factor? Several other factor may have changed from one year to another. Therefore, attributing the year effect to only precipitation would not be accurate. This could only be considered a “year effect”, which then authors may infer that rainfall was the main difference. But this would be only a potential explanation to discuss, nothing conclusive.

Given the many factors being tested and with potential interaction effects, the design and models used should account for this other sources of variation (double and triple interactions, maybe even quadruple interaction if the degrees of freedom of the model allows for its assessment).

Using contrast would be helpful to compare groups of treatments, e.g., not rested vs. rested.

You state that you used “descriptive statistics” (I assume this is shown in figure 6) but in the Results section you describe “overall” effects and ANOVA results. All statistical analysis section needs improvement and rephrasing.

L155: please replace “variables” by “factors”

Results

L162-164: remove this paragraph since it is redundant.

L168-169: you are presenting the main effect of “year”, therefore refer to the actual period describe (2017/2018 or 2018/2019). Do not use “favorable year” or climatic conditions” to refer to the particular years.

L172: how can an interaction effect be tested with one-way ANOVA? There is a flaw in the methodology. Please amend as suggested previously.

Figures 2 to 5 do not allow for visualising interaction effects. Please add captions for sub-figure a, b, c and d. Figures and tables should be self-explanatory, i.e., readers should be able to understand what is shown in the figure without the need to look for additional information in the main text. Please enhance the title of the figure to make sure it is fully clear what is the information presented in each one.

L201: what do the authors mean by “regardless if the year and rest period”? Does that mean that there was not interaction effect?

Figure 4 and 5: the letters for mean separation is missing. And this should be reflected in the description of results (authors compare minimum and maximum values but did not clarify whether intermediate values were different or not form the other means).

L217: what do the authors mean by “the combined effect of all tested factors”? Was that p value obtained from the ANOVA model of a fully factorial combination of all the factors or by introducing all the treatments combinations into a single one-way ANOVA? It seems the latter is true, and this is a huge statistical mistake. Moreover, should any of the interactions be significant, you can only describe the results in terms of the combination of treatments instead of only the main factors. If figure 6 is the results of a one-way ANOVA using all the groups, this is incorrect. Moreover, there are not letters for mean separation.

L227: “grazing parameter” is not a correct term for the indices. Please amend.

L229: please delete “ANOVA revealed a highly significant effect of rainfall amount and distribution”. The “ANOVA revealed” expression is redundant, and the remaining text is not accurate: you are testing the “year effect”. It should read: “However, there was an effect of year on N1…”.

L232: it’s not “climate”, it’s “year”.

L248-249: it should read “There was no interaction effect between the year and grazing exclusion on N2 (p = 0.112)”.

L253-255: any reference to the control treatment?

L262: Please delete “the ANOVA showed a highly significant” – this is redundant. This sentence should read “There was an effect of grazing exclusion in E20 (p<0.01)”.

L269: “The combined effect of all factors on …” – how was this done? Descriptive statistics? One-way-ANOVA? Both are incorrect.

L279: Please delete: “The simple linear regression model showed that” – this is redundant.

I have not revised the Discussion not the Conclusions sections since the previous sections will need to be improved and rewritten before the actual findings can be assessed.

Author Response

Review Report Form 2

General comment: 

The main flaw of the manuscript is the statistical approach and its description. The authors state that a one-way ANOVA was used to analyse the difference between main factors (grazing exclusion, rest duration, vegetation type and year effect). However, when presenting the results, they mention “interaction effects” – these effects can only be tested with various types pf factorial designs. Given that there are many factors being considered and several potential comparisons, the authors should seek for statistical advice on the appropriate experimental design and statistical model, and on the analysis to be performed (e.g. contrasts). Once the statistical approach is proved correct, the authors should amend the Results and Discussion section accordingly. Do not use “significant” or significantly” – that is redundant. You should not describe differences between groups if they were not statistically significant. Please present the actual p value (instead of “<0.05”). Please add lowercase letter to figures with more than two treatments to separate means. Do not use the term “parameter” interchangeably with “variable”. You are measuring variables (sample mean) to try to estimate parameters (population mean).

Response

This is a valid concern and we like to thank reviewer for raising this issue. The statistical analysis has been revised entirely taking into consideration all the comments provided by the estimated reviewer.

Specific comments:

Title: weather variation between two consecutive years is not climate change. The title should be amended to describe the actual work presented.

Response

The title has been modified taking into account reviewer suggestion

Comment

L43-56: please rephrased the sentence since it is not well structured.

Response

We would like to thank the reviewer; the paragraph is revised to be well structured.

Comment

“Continuous grazing” – the correct term is “continuous stocking” as per Allen et al. (2011) (doi: 10.1111/j.1365-2494.2010.00780.x). Please find a replace the term throughout the manuscript.

Response

We appreciate the reviewer efforts to provide the references of rangelands terminologies, however, not all the recommended terminologies are used in the international rangelands Journals  is using grazing not stocking, kindly check: https://www.publish.csiro.au/CP/CP19156;  https://www.sciencedirect.com/science/article/abs/pii/S1550742420300828; https://www.ncbi.nlm.nih.gov/pmc/articles/PMC7068843/

 Material and Methods

Comment

L76: “more than 1,349,410 head” – this looks like a very precise number to be referred as “more than”. It could be referred as “more than 1.3 million heads” to match the style for the area covered (“about 1.5 million ha”).

Response

We fully agree, the values area corrected as suggested by the estimated reviewer.

Comment

Figure 1 – what is the period for the “mean” series? Usually a climatic means is taken from 30 years of data. Also in this figure “annual and monthly” -  I can see only the monthly values. Please amend.

Response

The period for the mean is added. The word “annual” is deleted.

Comment

L85: “This study…” (delete “The”). Should we assume that each private rangeland contained all the four vegetation types? Or should we assume that each private rangeland had one only vegetation type? This is not clear at all. Please clarify it.

Response

‘The” is deleted.

The Experimental Design and Field Measurements section was revised entirely. We opted not to mention the land tenure to avoid any confusion. FYI, both communal and private range lands in southern Tunis are under the same kind of management, open for continuous grazing unless there is a contract with local authority (Government agency) their protection. In our experiment each selected site has one plant community except for the control which reflect the heterogeneous rangeland dominated by unpalatable and herbaceous species due to continuous grazing

Comment

L106: please state what are the spring months and how the plant communities were selected.

Response

The required information was added in the revised version. The plant communities mentioned in this study are the main ones present in southern Tunisia.

Comment

L118: what do the authors mean by “minimal area”? Please clarify it in the manuscript.

Response

The paragraph was revised

Comment

L123: “is equal to the number of points multiplied by 100” -  I would have thought a proportion is a number resulting from counting unit DEVIDED into 100. Please clarify or amend.

Response

Thank you very much for the comment. The correction was made and reflected in the text.

Comment

L125-128: these lines should be removed from this section since they are discussing the methodology. They could be moved to the Discussion section if they are relevant to discuss any of the findings.

Response

The Experimental Design, Field Measurements and Discussion sections have been revised.

Comment

L128-129: this may be rephrased as “Hill’s unified diversity indices N0, N1 and N2 [31] were calculated:”

Equations: only one equation for N1 should be presented – the authors have presented a kind of demonstration about why N1 equals H’. Please only use the one that uses p (in the second line of N1 equations shown) to match the style of the equation for N2.

Response

The Experimental Design, Field Measurements and Results sections were revised. The equations were also corrected

Comment

L139-146: these lines should be removed from this section since they are discussing the methodology. They could be moved to the Discussion section if they are relevant to discuss any of the findings. Please start this paragraph with “The relative proportion of dominant species…”

Response

The Experimental Design and Field Measurements section were revised.

Comment

Statistical analysis

How was the “private rangeland” considered into the model? Was is a block? Was is the “main-plot” (assuming a potential split-plot design)? Was it just another factor (in turn associated with the vegetation type)?

Response: As mentioned before we opted not to mention the land tenure to avoid confusion. Both communal and private range lands in southern Tunis are under the same kind of management, open for continuous grazing unless there is a contract with Government for protection. In our statistical model the grazing management was considered as a factor. The land ownership was mentioned when we first submitted manuscript to just describe the current rangelands situation in southern Tunisia

Comment

How was year included in the model? Just as a main factor (“year effect”) or as a blocking factor? Several other factor may have changed from one year to another. Therefore, attributing the year effect to only precipitation would not be accurate. This could only be considered a “year effect”, which then authors may infer that rainfall was the main difference. But this would be only a potential explanation to discuss, nothing conclusive.

Response: The year effect was considered as a factor. The Statistical Analysis section was revised completely.

Given the many factors being tested and with potential interaction effects, the design and models used should account for this other sources of variation (double and triple interactions, maybe even quadruple interaction if the degrees of freedom of the model allows for its assessment).

Using contrast would be helpful to compare groups of treatments, e.g., not rested vs. rested.

Response: The Statistical Analysis was revised completely. We used LSD to compare the treatments

You state that you used “descriptive statistics” (I assume this is shown in figure 6) but in the Results section you describe “overall” effects and ANOVA results. All statistical analysis section needs improvement and rephrasing.

Response: contrast would be helpful to compare groups of treatments, e.g., not rested vs. rested.

Response: The Statistical Analysis was revised completely. We used LSD to compare the treatments

L155: please replace “variables” by “factors”

Response: The Statistical Analysis was revised completely.

Results

L162-164: remove this paragraph since it is redundant.

Response: the paragraph is deleted.

L168-169: you are presenting the main effect of “year”, therefore refer to the actual period describe (2017/2018 or 2018/2019). Do not use “favorable year” or climatic conditions” to refer to the particular years.

Response: the Results section was revised entirely.

L172: how can an interaction effect be tested with one-way ANOVA? There is a flaw in the methodology. Please amend as suggested previously.

Response: the Statistical Analysis and results sections were revised completely.

Figures 2 to 5 do not allow for visualising interaction effects. Please add captions for sub-figure a, b, c and d. Figures and tables should be self-explanatory, i.e., readers should be able to understand what is shown in the figure without the need to look for additional information in the main text. Please enhance the title of the figure to make sure it is fully clear what is the information presented in each one.

Response: the Results section was revised entirely including the graphs.

L201: what do the authors mean by “regardless if the year and rest period”? Does that mean that there was not interaction effect?

Response: the section was revised entirely,

Figure 4 and 5: the letters for mean separation is missing. And this should be reflected in the description of results (authors compare minimum and maximum values but did not clarify whether intermediate values were different or not form the other means).

Response: the Results section was revised entirely including the graphs.

L217: what do the authors mean by “the combined effect of all tested factors”? Was that p value obtained from the ANOVA model of a fully factorial combination of all the factors or by introducing all the treatments combinations into a single one-way ANOVA? It seems the latter is true, and this is a huge statistical mistake. Moreover, should any of the interactions be significant, you can only describe the results in terms of the combination of treatments instead of only the main factors. If figure 6 is the results of a one-way ANOVA using all the groups, this is incorrect. Moreover, there are not letters for mean separation.

Response: the Results section was revised entirely including the graphs.

L227: “grazing parameter” is not a correct term for the indices. Please amend.

Response: this has been revised

L229: please delete “ANOVA revealed a highly significant effect of rainfall amount and distribution”. The “ANOVA revealed” expression is redundant, and the remaining text is not accurate: you are testing the “year effect”. It should read: “However, there was an effect of year on N1…”.

Response: The section is revised completely

L232: it’s not “climate”, it’s “year”.

Response: The section is revised completely

L248-249: it should read “There was no interaction effect between the year and grazing exclusion on N2 (p = 0.112)”.

Response: The section is revised completely

L253-255: any reference to the control treatment?

Response: The section is revised completely

L262: Please delete “the ANOVA showed a highly significant” – this is redundant. This sentence should read “There was an effect of grazing exclusion in E20 (p<0.01)”.

Response: The section is revised completely

L269: “The combined effect of all factors on …” – how was this done? Descriptive statistics? One-way-ANOVA? Both are incorrect.

Response: The section is revised completely

L279: Please delete: “The simple linear regression model showed that” – this is redundant

Response: The section is revised completely

Reviewer 3 Report

The article deals with important issues related to the preservation of natural ecosystems subjected to grazing management in the conditions of climate change. The research results, apart from theoretical significance, may have an impact on the optimization of the rangeland use. The conclusions suggest that the temporary grazing exclusion has a positive effect on the state of the ecosystem, expressed by the diversity indices.

However, higher number of species is not always beneficial for ecosystem.  Important is also what species start to grow. This should be also mentioned in the paper.

The paper needs significant improvement.

Materials and Methods

Lack of site location. Maps with study areas? Are the areas spaced apart? Are the amounts of rainfall not varied locally (one measurement site?). The precipitation data for both periods is much higher than the multiannual averages, can 2018/2019 be considered as typical?

The scheme of the experiment is very imprecisely described.

As I understand, permanent plots were selected in 2018 on areas not grazed for 1, 2 and 3 years.  If  lists of species were repeated in exactly the same permanent plots in 2019 why they were also selected in 2019 (line 106). Were the vegetation sampling repeated at exactly the same plots in both years? This will affect the statistical models. In which month survey was made? Why the minimum plot area of ​​256 m2?

Control plots with continues grazing should be in each type of vegetation.

The applied indices of diversity do not have to be accurately described with the help of detailed mathematical formulas. They are widely used. It is enough to give a general formula with reference to the source and, for example, their relationship to Shannon and Simpson indices. It cannot be written that N1 is the number of abundance and N2 of very abundance species. They include the abundance data of species, but this are not the species numbers. The evenness notation is E2,0.

In general statistical analyses are insufficiently describe. Did the data meet the conditions for using ANOVA (normal distribution?). It is worth providing the exact results of statistical analyzes in the table. Were the reported differences between e.g. communities analyzed statistically (eg. post-hoc test)?

The text (e.g. line 172) mentions the lack of interaction between the variables, but in the description of the analyzes there is only one-way Anova.

A lot of treatments were used in the analysis and only 3 repetitions. In the case of objects that are very diverse and located at a great distance from each other, the influence of various factors may be significant. If the censuses of vegetation in both years were made on permanent plots, it would be better to analyze the effect of the duration of exclusion from grazing for each type of vegetation separately.

Comparing each treatments with each other does not answer any question. It also makes no sense to compare the results with just one grazing area.

Correlations between indices between the indices are rather obvious and add nothing new. There are also not discussed.

Necessary linguistic correction. Some wording is inappropriate.

Grazing exclusion is better than 'rest' or 'protection'.

(table 1) Rangeland vegetation type rather than Community and why a 'targeted plant community'? Treatment code instead of Code. Add the year of research

line 54-57 incomprehensible sentence

line 60 rangeland health at critical time - rangeland condition? specify 'critical time'

line 66 years (2018 and 2019) with different climatic conditions

line 85 The or This

maybe add 'vegetation' (arid rangeland types)

line 162-164 is redundant

line 172 level of disturbance - what does it mean?

line 458 according to the methodology (Daget and Poissonet), numerical values ​​of the abundance of occurrence can be given

Lack of a, b, c, d explanation in table captions

Author Response

Review Report Form # 3

The article deals with important issues related to the preservation of natural ecosystems subjected to grazing management in the conditions of climate change. The research results, apart from theoretical significance, may have an impact on the optimization of the rangeland use. The conclusions suggest that the temporary grazing exclusion has a positive effect on the state of the ecosystem, expressed by the diversity indices.

However, higher number of species is not always beneficial for ecosystem.  Important is also what species start to grow. This should be also mentioned in the paper.

Response

We would like to thank the reviewer for his comments. We fully agree. However, in our research we recorded all the species we encountered in the four plant communities during the study. We provided the information about all available species based on their proportion in the plant community and their palatability is mentioned in appendix A. This information offers us an indication about the contribution and importance of each species to the ecosystem in general and for animal feeding in particular.

Comment

Lack of site location. Maps with study areas? Are the areas spaced apart? Are the amounts of rainfall not varied locally (one measurement site?). The precipitation data for both periods is much higher than the multiannual averages, can 2018/2019 be considered as typical?

Response

The map showing site location has been added. Since the target areas are not spread apart, the rainfall distribution is the same across all plant communities. However 2017/18 precipitation was higher than 2018/19.

Comment

As I understand, permanent plots were selected in 2018 on areas not grazed for 1, 2 and 3 years.  If  lists of species were repeated in exactly the same permanent plots in 2019 why they were also selected in 2019 (line 106). Were the vegetation sampling repeated at exactly the same plots in both years? This will affect the statistical models. In which month survey was made? Why the minimum plot area of ​​256 m2?

Control plots with continues grazing should be in each type of vegetation.

The applied indices of diversity do not have to be accurately described with the help of detailed mathematical formulas. They are widely used. It is enough to give a general formula with reference to the source and, for example, their relationship to Shannon and Simpson indices. It cannot be written that N1 is the number of abundance and N2 of very abundance species. They include the abundance data of species, but this are not the species numbers. The evenness notation is E2,0.

Response

The Experimental Design and Field Measurements sections were revised entirely taking into consideration all the comments provided by the reviewer.

Comment

line 54-57 incomprehensible sentence

Response

The introduction section was enhanced thanks to the comments made

Comment

line 60 rangeland health at critical time - rangeland condition? specify 'critical time'

Response

The introduction section was revised completely and the 'critical time' is deleted

Comment

line 66 years (2018 and 2019) with different climatic conditions

Response

The section is modified we used: “and climatic variability (2018 and 2019)?”

Comment

line 85 The or This, maybe add 'vegetation' (arid rangeland types)

Response

“The” is deleted

Round 2

Reviewer 2 Report

Thanks to the authors for addressing most of my previous comments. The manuscript has improved in some aspects. However, relevant aspects are not clear or seem to be deficient, particularly the experimental design and statistical analysis.

The authors state that (L13-15 and L169-170) that the design was fully factorial. However, it is not clear what is the role of the “Control” group within the design. Sometimes is treated as a plant community (in that case it should have been subjected to the same factors as the plant communities, i.e., year and resting period) and sometimes is treated as a resting regime (in that case there should have been one Control treatment in each plant community). I cannot se how the Control was included in the fully factorial design.

There is lack of information on the number of replicates: 3 transects within one only field is not a real replicate but a pseudo-replicate, which does not allow for statistical comparisons. It is not clear either how many sites were used for the study and where exactly they were. It also seems that each plant community was associated with one soil type. Is that correct? According to authors response to my initial report, each plant community was found in only one site. If so, how can the authors attribute the results to the plant community instead of to the specific site with its specific characteristics?

The authors insist to call the year effect as the “climatic conditions” or the “rainfall regime”. They may provide potential explanations for the differences observed in the variable analysed between years to the amount of rainfall, but they cannot call the year effect  the “rainfall” or “climatic” effect. You would need a longer time series to attribute the variation between years to the rainfall differences. There are other weather variables that could differ between years and they were not included in the study.

The p values for the double and triple interactions should be included. A table with the ANOVA results would be very useful. As a general rule, when an interaction effect is found, the results must be discussed in relation to the combinations of the levels of factors and not at the main factor level. Therefore, Figure 2 is appropriate (although it lacks of the letters indicating mean separation) but Figure 3 (there are two figures 3!), Figure 4 (the second Figure 3), and Figure 5 are not appropriate (the show main factors or a partial combination of them, however there was a triple interaction effect).

L226-227: does this mean that an interaction effect between plant community and resting period was found? What about the trippel interaction with year? Figure 6 lacks of letter for mean separation.

L263-266 and Figure 8 should not be included since there were not differences between resting periods.

What is the relevance of presenting the correlation between indices?

Why the authors call 2018/2019 a “normal” year when there was a 50% higher rainfall than the 20-y average (climatic averages are made of 30 years!).

“Parameter” is different than “variable”.

L412-418: The authors do not know what would happened in a “normal” year with 1 year of grazing. What would be the recommendation then?

L-420-423 and 436-440 are not part of the Conclusions section.

Author Response

Review # 2 comments:

thanks to the authors for addressing most of my previous comments. The manuscript has improved in some aspects. However, relevant aspects are not clear or seem to be deficient, particularly the experimental design and statistical analysis.

Response: We would like the reviewer for his valuable comments which resulted in the manuscript improvement

The authors state that (L13-15 and L169-170) that the design was fully factorial. However, it is not clear what is the role of the “Control” group within the design. Sometimes is treated as a plant community and sometimes is treated as a resting regime (in that case there should have been one Control treatment in each plant community). I cannot see how the Control was included in the fully factorial design.

Response: The control site in this study represents a large area which is not protected (open to grazing) as it illustrates the common practice as done by the farmers in the region. In addition, due to its extend, the control site was located adjacent to the four plant communities. Unfortunately, because this area has been subjected to continuous grazing for many years, there is no clear presence of a particular plant community (no dominant species) showing an advanced degree of degradation.

For the reasons listed above, we were not able to identify a control for each plant community and even if we did there would not be any differences among the controls as the entire site is basically covered with bare soil.

There is lack of information on the number of replicates: 3 transects within one only field is not a real replicate but a pseudo-replicate, which does not allow for statistical comparisons. It is not clear either how many sites were used for the study and where exactly they were. It also seems that each plant community was associated with one soil type. Is that correct? According to authors response to my initial report, each plant community was found in only one site. If so, how can the authors attribute the results to the plant community instead of to the specific site with its specific characteristics?

Response: To avoid pseudo-replication, we have stated that the transects and the samples taken inside the quadrats in each site were considered as replicates.

With respect to the second part of the comment, we would like to clarify some points: the plant community is identified based on the dominant species composition in a specific area. No doubt interactions among plant species with soil properties and other environmental variables is well known and documented (Bagheri et al., 2017; Mirzaei Mossivand et al., 2017). In our case, the four plant communities are located in the same agroecological boundary (10 km2) under the same climatic conditions (rainfall, temperature, humidity). Thus, the focus of this study is on plant communities rather than location, since each community is characterized by its own soil type. In our discussion, we clearly associated these specifications to our results.

The authors insist to call the year effect as the “climatic conditions” or the “rainfall regime”. They may provide potential explanations for the differences observed in the variable analysed between years to the amount of rainfall, but they cannot call the year effect the “rainfall” or “climatic” effect. You would need a longer time series to attribute the variation between years to the rainfall differences. There are other weather variables that could differ between years and they were not included in the study.

Response: We agree with the reviewer’s suggestions. We changed the climatic conditions to year effect.

The p values for the double and triple interactions should be included. A table with the ANOVA results would be very useful. As a general rule, when an interaction effect is found, the results must be discussed in relation to the combinations of the levels of factors and not at the main factor level. Therefore, Figure 2 is appropriate (although it lacks of the letters indicating mean separation) but Figure 3 (there are two figures 3!), Figure 4 (the second Figure 3), and Figure 5 are not appropriate (the show main factors or a partial combination of them, however there was a triple interaction effect).

Response: As suggested by the reviewer, the ANOVA table is provided, and the Figures are adjusted accordingly.

L226-227: does this mean that an interaction effect between plant community and resting period was found? What about the Trippel interaction with year? Figure 6 lacks letter for mean separation.

Response: The sentence was modified for more clarity. The graph format has been revised.

L263-266 and Figure 8 should not be included since there were not differences between resting periods.

Response: the graph was removed as suggested by the reviewer.

What is the relevance of presenting the correlation between indices?

Response: We believe correlation between indices provides insight and summarizes the relationship between them

Why the authors call 2018/2019 a “normal” year when there was a 50% higher rainfall than the 20-y average (climatic averages are made of 30 years!).

Response:  We would like to thank the reviewer for his comment. Depending on the information of the local pastoralists and rangeland researchers, rainfall amount and rainfall distribution are considered among the most important factors that affect the biomass production of the range lands in Tataouine.  From their experience when the total of 60 mm rainfall occurs during the September to December months this is a valid indicator to have normal biomass production of the rangelands in the studied area.  While when the amount exceeds 80 mm this will lead to good year rangeland production.  However, when the annual rainfall is less the biomass will be less.

“Parameter” is different than “variable”.

Response: This was corrected.

L412-418: The authors do not know what would happen in a “normal” year with 1 year of grazing. What would be the recommendation then?

Response: Keeping in mind that favorable years are the exception and normal or drought years are the default. During normal year the clauses related to livestock exclusion remain the same which means no grazing is allowed.

When we designed our research, the main objective was to evaluate the duration of resting on the plant communities. However, the effect of an exceptional year was obvious, and it has significantly affected the results. Therefore, from our observations many pastoralists who were committed to protect their rangelands  for three years, they have decided to use the abundant biomass production resulted from the exceptional rainfall which was not the case of one year protection during a normal year. Therefore, we did not recommend to open grazing during normal years which means the site remain protected as agreed in the contract signed between the governmental agency and the pastoralist.

L-420-423 and 436-440 are not part of the Conclusions section.

Response: We appreciate the reviewer’s feedback. We made slight changes to the wording. We believe the conclusion should summarize the reason(s) why this study was conducted and the main take home messages for the broad readers. In particular what are the key implications for management?

Reviewer 3 Report

Comments in an enclosed file

Author Response

Review # 3 comments:

The authors made a lot of progress in improving the article, but some corrections regarding statistical analysis and editing errors are still necessary.

The main objection is the lack of presentation of the statistically significant differences between the objects. The authors write about the use of Fisher's Least Significant Difference (post-hoc test), but do not include this in the results. The statistical significance of the interaction of factors with communities is also not provided. 

Response: we would like to thank the reviewer for his comments. The ANOVA table is added.

With respect to the LSD presentation in majority of the graphs two direction standard error bars.

“The LSD can be replaced by the standard error and the rule of thumb that the LSD is approximately three standard errors: Hall, J.W. 1997. The presentation of statistical results in journal articles. Can J. Plant Sci. 77:11–14.”

Having said the letter is added to the graphs

Other remarks

It is necessary to standardize the type of charts. Charts 5 and 6 should be of the same type, as well as charts 2, 3 and 7. Why are some graphs presented as the bar type and the other as  box-whisker plots?

Fig. 6 why Control for one year only?

Response: The control site for the plant community was the site where there is no dominant species this mainly to the degradation level resulted from miss-management. Whereas the control site of the resting regime is a site where the it was freely grazed for more than three. With respect to the reviewer suggestion “(in that case there should have been one Control treatment in each plant community)” the control is an area where there is no plant community. We don’t think there is a point to have one control for each plant community as there will not be any differences in the three controls as the four communities are located in the same site and the areas where there is no plant communities are having the same species.

For ease of comparison, Control should be at the beginning of the axis in the figures.

Response: The graphs were changed.  

line 209 Figure 4

Response: The number was changed.  

line 227 communities

Response: The word was changed.  

line 252 varied significantly – unnecessary

Response: “varied significantly” was deleted   

line 319-321 “the results of many studies that showed an increase of this parameter with disturbance and vice versa”.  Vice versa means increase disturbance with increase of N0 

Response: “Vice versa” was deleted   

line 352 mobile sand – is this correct ?

Response: “mobile sand” was changed to sand dunes

 line 408 social fencing – is this correct?

Response: “social fencing” was changed to based on social awareness

Round 3

Reviewer 2 Report

Thanks for clarifying the design and improving the description of the analysis.

Author Response

Comments:

Thanks for clarifying the design and improving the description of the analysis.

Response:  We would also like to take this opportunity to express our thanks to the reviewer for the positive feedback.